# Unifying Spike Perception and Prediction: A Compact Spike Representation Model using Multi-scale Correlation

## ABSTRACT

The widespread adoption of bio-inspired cameras has catalyzed the development of spike-based intelligent applications. Despite its innovative imaging principle allows for functionality in extreme scenarios, the intricate nature of spike signals poses processing challenges to achieve desired performance. Traditional methods struggles to deliver visual perception and temporal prediction simultaneously, and they lack the flexibility needed for diverse intelligent applications. To address this problem, we analyze the spatio-temporal correlations between spike information at different temporal scales. A novel spike processing method is introduced for compact spike representations that utilizes intra-scale correlation for higher predictive accuracy. Additionally, we propose a multi-scale spatio-temporal aggregation unit (MSTAU) that further leverages inter-scale correlation to achieve efficient perception and precise prediction. Experimental results show noticeable improvements in scene reconstruction and object classification, with increases of **3.49dB** in scene reconstruction quality and **2.20%** in accuracy, respectively. Besides, the proposed method accommodate different visual applications via switching analysis models, offering a novel perspective for spike processing.

## CCS CONCEPTS

• **Computing methodologies** → **Computer vision**.

## KEYWORDS

spike processing, perception and prediction, multi-scale aggregation, scene reconstruction, object classification

## 1 INTRODUCTION

With the popularization of autonomous driving and Industry-4.0, an innovative biologically inspired spike camera has emerged, which emulates the fovea of mammalian retina [31] [3]. Breaking the limitation of exposure [18], the spike camera emits spikes at each position asynchronously and achieves a maximum temporal resolution of up to 40k frames per second (FPS). This can effectively mitigate motion blur and content missing, supporting visual tasks in scenarios with high-speed motion, such as scene reconstruction and classification for fast moving objects. In practical usages, intelligent applications are usually deployed in an end-cloud collaborative architecture due to the weak computational ability of

*ACM MM, 2024, Melbourne, Australia*

© 2024 Copyright held by the owner/author(s). Publication rights licensed to ACM.
ACM ISBN 978-x-xxxx-xxxx-x/YY/MM
https://doi.org/10.1145/nnnnnnn.nnnnnnn

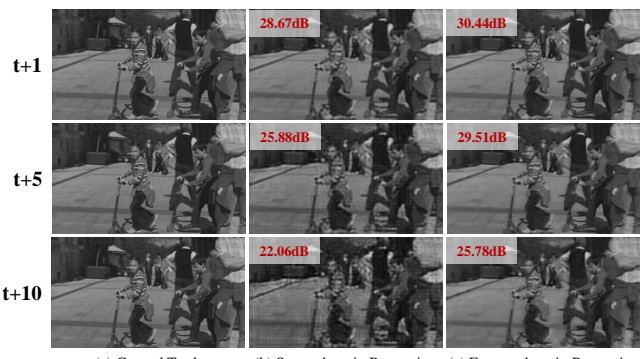

|  | (a) Ground Truth | (b) Scene-domain Processing | (c) Feature-domain Processing |

**Figure 1: Visual comparison scene reconstructed through scene-domain and feature-domain processing. Zoom for better visibility.**

cameras [38] [25]. The camera captures information and emits spikes, which are compressed and transmitted to a cloud server for further processing according to specific task. On the one hand, it is essential to extract sufficient semantic information from spike sequences to satisfy the precision requirements of visual applications at different granularities [23] [13] [16]. On the other hand, accurate prediction of future spatio-temporal states is also crucial for proactive decision-making ahead of time [21] [20]. This imposes demands on perception and prediction of spike signals simultaneously.

Given the analyzed results, an intuitive strategy for simultaneous perception and prediction of spike signals is to first process spikes to meet demands of visual applications and then perform temporal predictions on the result sequences. However, this struggles to effectively address these challenges. *On one hand, although this approach can achieve satisfactory performance in specific task, it lacks versatility for scenarios with multiple tasks, especially when these applications involve different semantic levels [36] [37]. On the other hand, inconsistency between the target of perception and prediction lead to distortion, which accumulates gradually over temporal interval, making it increasingly difficult to effectively perceive after predictions [15].* For instance, in scene reconstruction, spike sequence can be rebuilt into a series of scenes initially, upon which video prediction methods are applied for forecasting. However, results in Fig. 1(b) show that the predictive quality is less efficient enough. One issue is that the reconstructed scenes exhibit severe distortion compared with ground truth, which escalates with the extension of prediction interval. Moreover, prediction in pixel-level accuracy is limited so that the abundant temporal information contained in the spike sequence is used to compensate for the lack of spatial information (e.g., calculating the average firing rate and estimating the light intensity at the current moment based on spike generation over a period of time). In contrast, prediction on scenes rebuilt from spikes requires joint modeling of spatio-temporal information from historical observations. With temporal details discarded in reconstructed scenes and only spatial information preserved, achieving

high-accuracy forecasts becomes challenging. Besides, this method is not directly applicable for other visual tasks and retraining is required for all modules. Therefore, there is an urgent necessity to propose an effective method for spike processing with perception and prediction.

Building upon the above discussion, we introduce an innovative spike processing method considering multi-scale correlations, which are categorize as intra-scale and inter-scale one. Compared with previous approaches, information within spikes can be predicted in temporal domain with higher efficiency and fidelity, which can further serve for downstream applications with better performance. The major contributions are summarized as follows:

- We present a novel spike data processing method utilizing compact spike representations through multi-scale correlations. Our method offers perception and prediction at multiple semantic levels, providing a fresh perspective on spike visual intelligence.
- Leveraging strong stability within temporal scale, we propose an Intra-scale Correlation based Prediction (ICP) approach to learn from the Multi-scale Temporal Filters (MTFs) for compact spike representations. Moreover, we further consider inter-scale correlation and introduce a Multi-scale Spatio-Temporal Aggregation Unit (MSTAU) for the joint modeling of spatio-temporal information across different scales, enabling efficient spike representation prediction. The predictive represents are adaptable for low-level and high-level spike visual tasks simultaneously.
- Extensive experimental results demonstrate that our method achieves a significant improvement for predictive accuracy of spike perception, compared to previous SOTA methods. In terms of downstream applications, we achieve a noticeable increase of **3.49dB** in PSNR for scene reconstruction and an improvement of **2.20%** in Top-1 accuracy for classification, setting a robust foundation for spike processing.

## 2 RELATED WORK

Scene reconstruction and object classification are two important visual applications in spike processing. The former utilizes the principle of spike firing, which is stimulated by luminance intensity, to rebuild scenario content with high pixel-level fidelity. The latter involves categorizing objects captured by spike cameras based on higher-level semantic information, resulting in superior accuracy.

**Scene reconstruction.**    Chen *et al.* [28] introduced a tokenizer and spatio-temporal attention significantly enhances the accuracy and stability of reconstruction quality. Zhu *et al.* [42] put forward a framework for retina-like visual image reconstruction, harnessing bio-realistic Leaky Integrate and Fire (LIF) neurons along with synapse connections governed by spike-timing-dependent plasticity (STDP) rules. Additionally, Spk2ImgNet proposed in [39] stood out for its ability to reconstruct dynamic scenes from continuous spike streams through the use of deep learning techniques. Wei *et al.* [19] also introduces a retinal spike train decoder to enhance the accuracy of reconstructing visual scenes from retinal spike trains, demonstrating its potential to advance brain-machine interface technologies by improving how visual scenes are decoded from spike data. Xiang *et al.* [35] contributed to the field by proposing

a learning-based super-resolution reconstruction method tailored for high temporal resolution spike streams. [8] presented a novel coarse-to-fine method utilizing region-adaptive-based spike distinction, which effectively differentiates between dynamic and static spikes to enhance the reconstruction of natural scenes.

**Object classification.**    Felix *et al.* [10] employed linear filters for enhanced signal representation and signal-to-noise ratio, and used well-defined thresholds for simultaneous spike detection and classification. Wilson *et al.* [33] utilized multiple monotonic neural network to group spikes automatically via hierarchical clustering, which is visually compared with hand marked grouping on a single record. Recent years, Zhao *et al.* [40] introduced a modeling algorithm to assess the detection capability of spiking cameras under various scenarios, such as different brightness intensities and camera lens configurations. This algorithm helped determine appropriate camera settings for capturing high-speed objects. SpiReco mentioned in [41] discussed the potential of spike cameras in high-speed object recognition. The study highlighted the advantages of spike cameras, such as their high temporal resolution and dynamic range, while also acknowledging the challenges posed by their physical limits. Besides, a accelerator was implemented on FPGA for high-speed moving objects detection and tracking with a spike camera. This research explored hardware acceleration techniques to enhance the real-time processing capabilities of spike cameras, enabling them to detect and track high-speed objects more effectively.

## 3 METHODOLOGY

Building upon the discussion in Sec. 1, we propose to convert the spike sequence $\{S_t\}$ into compact spike representations $\{F_t\}$ according to the biological visual mechanism, which are rich in spatio-temporal information. Based on observations of historical representations $\{F_{t-i}\}$, we predict the representation at the current moment $\hat{F}_t$ and utilize it to generate the result $\hat{R}_t^V$ for applications, formulating as

$$F_t = \mathbb{FE}(Spk_{\{t-\Delta t:t+\Delta t\}}), \tag{1}$$

$$\hat{F}_t = Pred(F_{\{t-\tau:t-1\}}), \tag{2}$$

$$\hat{R}_t^V = \mathbb{A}^V(\hat{F}_t), \tag{3}$$

where $\mathbb{FE}$ and $Pred$ denote the feature extraction and prediction module respectively, and $\mathbb{A}^V$ is the analysis component for task $V$. On the one hand, $\{F_t\}$ are profuse with spatial information at different granularities, which can be reused for multiple downstream tasks at various semantic levels through different $\mathbb{A}^V$. On the other hand, $\{F_t\}$ are abundant with temporal information at different scales, enabling efficient and accurate forecast at multiple extents. Moreover, the dimensions of $\{F_t\}$ can be compacted to enhance the expressive capability of $\{F_t\}$ through increasing the density of information, while the computational complexity can be decreased in the meanwhile. Consequently, the compact representation-based spike processing approach can efficiently meet the demands of various applications and achieve high-precision temporal predictions.

### 3.1 Necessity of Multi-scale Temporal Filters

As spike cameras are bio-inspired, the processing for spike signals should also mimic biological visual mechanism to extract features

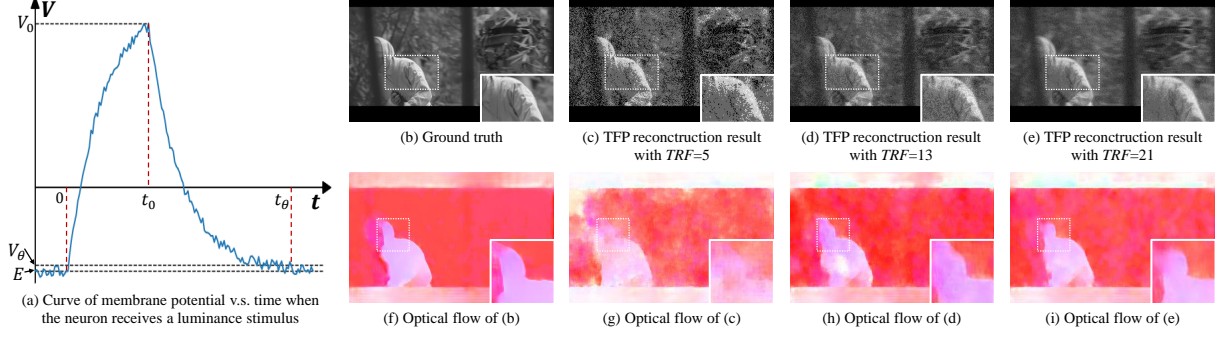

(a) Curve of membrane potential v.s. time when the neuron receives a luminance stimulus

(b) Ground truth

(c) TFP reconstruction result with *TRF*=5

(d) TFP reconstruction result with *TRF*=13

(e) TFP reconstruction result with *TRF*=21

(f) Optical flow of (b)

(g) Optical flow of (c)

(h) Optical flow of (d)

(i) Optical flow of (e)

**Figure 2: The neural dynamics curve stimulated by luminance and the spatio-temporal content comparison corresponding to different TRFs. Results illustrate a strong correlation between $t_\theta$ and $\nu$, while the information extracted by different TRFs has significant differences, demonstrating the necessity of utilizing MTFs to comprehensively extract information from spikes.**

abundant with spatio-temporal information [17] [2]. We decouple the persistent light stimuli into a sequence of transient luminance stimulus, each of which lasts for an extremely short time and thus the luminance can be approximated as constant. Taking the LIF as an example of neural dynamics [34] [14], for a neuron with time constant $\nu$, under the influence of transient luminance stimulus $I$ defined by

$$I = \begin{cases} I_0, & 0 < t < t_0, \\ 0, & else \end{cases}, \qquad (4)$$

we derive the temporal extent of this impact reaching up to $t_\theta$.

The entire process can be divided into two dependent procedures, which are the charging during $[0, t_0)$ and the discharging during $[t_0, t_\theta]$. The schematic diagram of the membrane potential changing over time is shown in Fig. 2(a). The neuro-dynamic differential equation can be expressed as

$$\nu \frac{\mathrm{d}V_t}{\mathrm{d}t} = -(V_t - E) + \alpha RI, \qquad (5)$$

where $V_t$ and $E$ denote the membrane potential at moment $t$ and the resting potential, $R$ is the membrane resistance and $\alpha$ represents the photovoltaic conversion efficiency [4] [27]. By joint solving equations for both two processes, $t_\theta$ can be formulated as

$$t_\theta = t_0 + \nu \ln \frac{\alpha RI_0 [1 - \exp(-t_0/\nu)]}{V_\theta - E}, \qquad (6)$$

where $V_\theta$ illustrates the threshold that can distinguish between electrical signals or disturbances. Noticing that $t_0$ is an infinitesimal, which means the equation above can be approximated via Taylor expansion as

$$t_\theta = \nu \ln \frac{\alpha RI_0 t_0}{(V_\theta - E)\nu} = \nu \ln \frac{\alpha RP_0}{(V_\theta - E)\nu}, \qquad (7)$$

where $P_0$ denotes the amount of photons received by the neuron per unit time [22]. For any moment $t$, $V_t$ is a nonlinear superposition of the effects of all luminance stimuli during $[t - t_\theta, t]$. After substituting the relevant parameters, the PLCC (Pearson Linear Correlation Coefficient) between $t_\theta$ and $\nu$ reaches **0.995** [7], indicating that their correlation is extremely high. Since the mammalian visual system consists of neurons with different time constants [11] [1] [24], it is essential to take into account the impact of various $\nu$ on neural dynamics in a comprehensive manner. [1]

---

[1]Detailed derivations are shown in the supplementary material.

From perspective of implementation, MTFs with various temporal receptive fields (TRFs) are required as a combination to extract features with different temporal granularities, which are subsequently associated to support intelligent applications. We further illustrate the necessity of using MTFs through a scene reconstruction task in both spatial (Fig. 2(b~e)) and temporal (Fig. 2(f~i)) domain. As for spatial texture, reconstructed results are significantly influenced by the stochastic nature of spikes (pixel-level noise) if TRF is too small, while an excessively large TRF makes it challenging for alignment, leading to severe motion blur. In terms of temporal characteristic, movement exhibits strong disorder and randomness for small TRF. In contrast, motion is more consistent and the boundaries between foreground and background are more pronounced for large TRF.

We assert that features produced by filters with varying TRFs exhibit unique temporal scales. Therefore, composing features $f_t^{1:K}$ at different scales into $F_t$ is crucial to maintain the richness and continuity of spatio-temporal information. Eq. 1 can be adapted as

$$f_t^{\{1:K\}} = \mathbb{MFE}(Spk_{\{t-\Delta t:t+\Delta t\}}), \qquad (8)$$

$$F_t = \mathbb{FC}(f_t^{\{1:K\}}), \qquad (9)$$

where $\mathbb{MFE}$ and $\mathbb{FC}$ denote the multi-scale feature extraction and feature composition module respectively, and $K$ represents the number of temporal scales.

## 3.2 Intra-scale Correlation based Prediction

To explore the discrepancies and correlations between features at different scales, we statistically analyze the spatio-temporal characteristics through TFP reconstruction results with various TRFs [26] [32], as shown in Fig. 3(a). The pixel smoothness in spatial domain is negatively correlated with TRF, while the motion intensity in temporal domain is positively associated with TRF. This strong correlation indicates that visual information corresponding to different TRFs have significant disparities. Therefore, the representation composed from features with varying scales possess more complex spatio-temporal characteristics, making it challenging to be precisely predicted.

Despite the different texture contents and motion trends extracted from features at various temporal scales, the spatio-temporal information within the same scale exhibits significant stability in temporal domain, referred as **intra-scale correlation**. We calculate

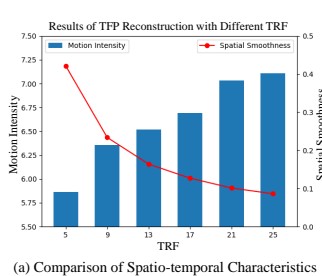
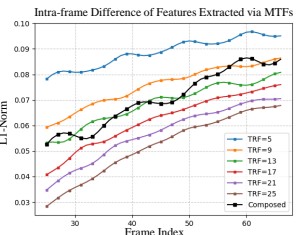

(a) Comparison of Spatio-temporal Characteristics between Different TRFs for TFP Reconstruction

(b) Comparison of Pixel-wise Differences between Adjacent Frames for Features Extracted via MTFs

**Figure 3: Comparison of spatio-temporal characteristics of features corresponding to different TRFs. Though there are significant variations between scales, temporal continuity within each scale is robust to facilitate accurate prediction.**

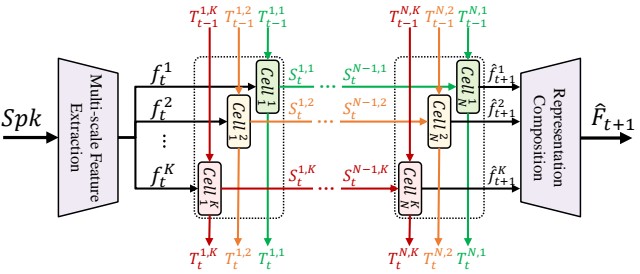

**Figure 4: Pipeline of prediction with ICP method. Leveraging intra-scale correlations, ICP predict features at each scale and ultimately combines them to form compact representations to support intelligent visual applications.**

the L1-norm of the residual between adjacent frames for features at each scale [12], as shown in Fig. 3(b). Results indicate that features at the same scale demonstrate strong temporally continuity with fewer noticeable fluctuations, denoting stable characteristics within scale. In contrast, although the representation is composed from these features, the emphatic temporal randomness and volatility make it challenging for prediction. Therefore, we propose an ICP method to model spatio-temporal characteristics of features at each scale separately for higher prediction accuracy, pipeline of which is illustrated as Fig. 4. With historical observations from last $\tau$ moments, each feature is predicted separately as $\hat{f}_t^k$, all of which are then composed as $\hat{F}_t$. Therefore, Eq. 9 and Eq. 2 can be adapted as

$$\hat{f}_t^k = Pred(f_{\{t-\tau:t-1\}}^k), \tag{10}$$

$$\hat{F}_t = \mathbb{FC}(\hat{f}_t^{\{1:K\}}), \tag{11}$$

where $f_{\{t-\tau:t-1\}}^{\{1:K\}}$ are historical features extracted according to Eq. 8.

## 3.3 Multi-scale Spatio-Temporal Aggregation Unit

Although there is a wide variation between features with different scales shown in Fig. 3(b), we observe a relatively consistent trend in their changes. Therefore, we calculate the PLCC between optical flows at various scales, as shown in Table. 1. Results indicate the existence of correlations between temporal scales, primarily occurring with larger TRFs and close TRF pairs (9&11 and 11&13). This relevance can serve as guidance information to supervise multi-scale

| TRFs | 7 | 9 | 11 | 13 |
|------|-----|-----|-----|-----|
| 7 | 1.0 | - | - | - |
| 9 | 0.0991 | 1.0 | - | - |
| 11 | 0.2063 | 0.5978 | 1.0 | - |
| 13 | 0.1026 | 0.1484 | 0.4367 | 1.0 |

**Table 1: PLCC between contents corresponding to different TRFs. Results indicate the existence of high correlations between larger TRFs and close TRF pairs (9&11 and 11&13).**

joint prediction, referred as **inter-scale correlation**. Therefore, we propose a Multi-scale Spatio-Temporal Aggregation Unit (MSTAU) considering both intra-scale and inter-scale correlations to predict features with high precision to support visual applications, the pipeline of which is depicted in Fig. 5(a).

At the $n^{th}$ layer and $k^{th}$ temporal scale, $MSTAU_n^k$ takes several spatio-temporal states as input, including:

- Spatial states of all $K$ scales from previous $\tau$ and the current time steps in the $n-1^{th}$ layer $S_{\{t-\tau:t\}}^{n-1,\{1:K\}}$.
- Temporal states of the current scale from previous $\tau$ time steps in the $n^{th}$ layer $T_{\{t-\tau:t-1\}}^{n,k}$.

$MSTAU_n^k$ outputs the spatial and temporal states of the current scale at the current time step $S_t^{n,k}$ and $T_t^{n,k}$ respectively, propagating to other MSTAUs. $N$ layers of MSTAU are stacked hierarchically to enhance the expression and prediction capabilities. Features $f_t^{\{1:K\}}$ are regarded as spatial inputs of the first layer $S_t^{0,\{1:K\}}$, while spatial outputs of the final layer $S_t^{N,\{1:K\}}$ are considered as prediction of features $\hat{f}_{t+1}^{\{1:K\}}$. As shown in Fig. 5(b), each MSTAU consists of the following three modules, which are coupled tightly to enhance predictive capability.

**Temporal Regression Module.** Based on the temporal relevance at each scale, this module regresses historical states towards the current moment and generate multiple regression results corresponding to each scale. Prior works transferred the relevance between $(t-\tau \sim t-1)$ and $t$ in the $n-1^{th}$ layer to the $n^{th}$ layer to predict the state at $t$ [6] [5]. However, these approaches performed poorly due to the lack of consideration for intra-scale and inter-scale correlations. Our module extends this concept to multiple scales, transferring the temporal relevance of all scales in the $n-1^{th}$ layer to the current scale in the $n^{th}$ layer for regression. For scale $\kappa$, similarities between spatial states at $t$ and $t-i$ in the $n-1^{th}$ layer is calculated as guidance on regression of temporal states, formulating as

$$q_i^{\kappa} = \sum_{(x,y)} [S_{t-i}^{n-1,\kappa} \odot (W_a * S_t^{n-1,\kappa})], \tag{12}$$

$$T_{reg}^{n,\kappa} = \sum_{i=1}^{\tau} T_{t-i}^{n,k} \cdot \frac{\exp(q_i^{\kappa})}{\sum_{j=1}^{\tau} \exp(q_j^{\kappa})}, \tag{13}$$

where $q_i^{\kappa}$ is the similarity score and $T_{reg}^{n,\kappa}$ denotes the regression result at scale $\kappa$ in the $n^{th}$ layer, which is regarded as a long-term regression for temporal state at moment $t$.

**Multi-scale Aggregation Module.** The major challenge addressed by this module is how to adaptively aggregate long-term regressions $T_{reg}^{n,\{1:K\}}$ corresponding to various scales and combine

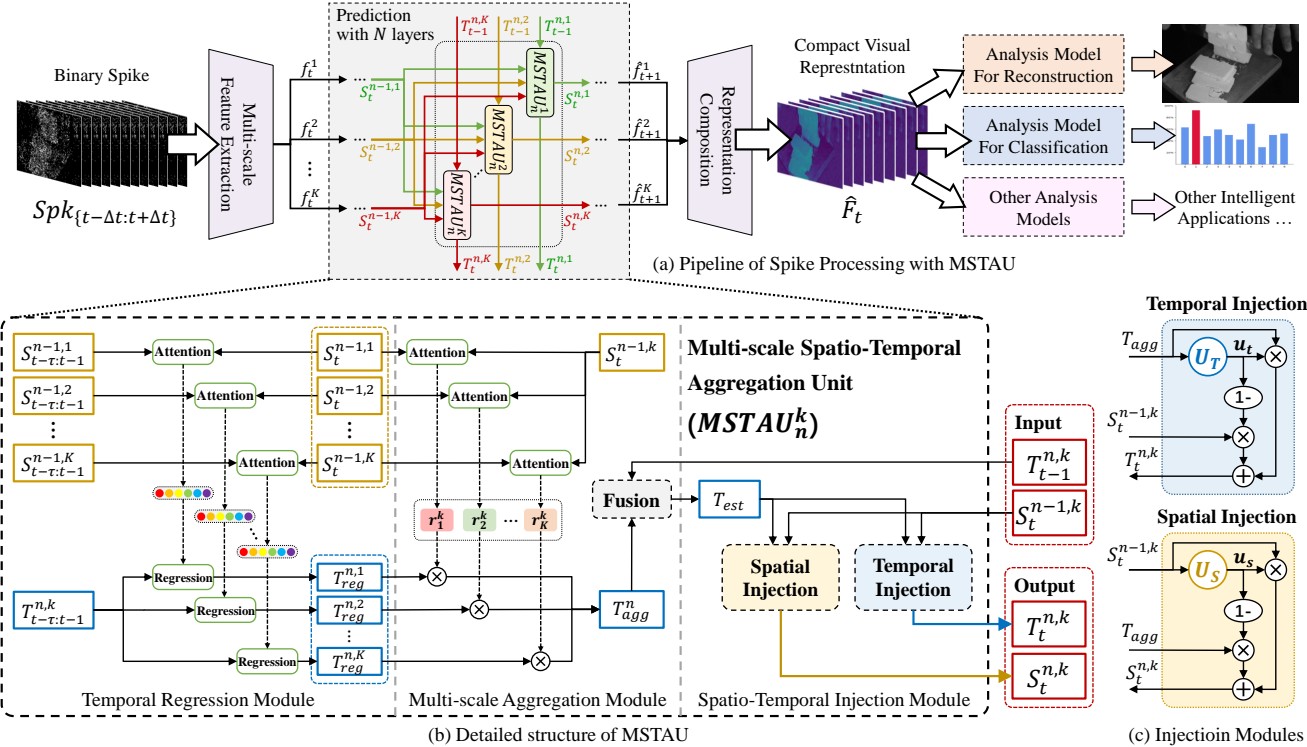

**Figure 5: The spike processing pipeline with MSTAU and the detailed structure of MSTAU and its sub-modules. By introducing inter-scale correlations to achieve efficient aggregation of features at different scales, MSTAU can propagate spatial and temporal information to other units, ultimately achieving high-precision prediction. Zoom for better visibility.**

them with momentary state $T_{t-1}^{n,k}$ to obtain the most accurate estimation of the temporal state. As for long-term aggregation, we transfer the correlation between all scales and the current scale from spatial states to temporal ones. Attention scores within scales $r_{\{1:K\}}^k$ are generated according to pixel-wise similarity, which then serve as navigation for fusion of $T_{\{t-\tau:t-i\}}^{n,k}$, formulating as

$$r_\kappa^k = \sum_{(x,y)} [S_t^{n-1,\kappa} \odot (W_a' * S_t^{n-1,k})], \quad (14)$$

$$T_{agg}^n = \sum_{\kappa=1}^{K} T_{reg}^{n,\kappa} \cdot \frac{\exp(r_\kappa^k)}{\sum_{\kappa=1}^{K} \exp(r_\kappa^k)}, \quad (15)$$

where $T_{agg}^n$ is respected as the comprehensive long-term aggregation for temporal state. As for long and short-term combination, $T_{t-1}^{n,k}$ is utilized for providing momentary information and maintaining content consistency. We employ an adaptive gating mechanism that generates fusion ratio, based on which to fuse trending and momentary information, formulating as

$$u_f = U_F(T_{t-1}^{n,k}) = \sigma(W_f * T_{t-1}^{n,k}), \quad (16)$$

$$T_{est} = u_f \odot T_{t-1}^{n,k} + (1 - u_f) \odot T_{agg}^n, \quad (17)$$

where $u_f$ represents the ratio generated from fusion gate $U_F$ and $T_{est}$ denotes the estimated temporal state.

**Spatio-Temporal Injection Module.** During information propagation, spatio-temporal states should be processed with injection bidirectionally to ensure the consistency. Drawing inspiration from the dual-gate mechanism, two gates $U_S$ and $U_T$ are employed to

generate injection ratios respectively, guiding the bi-directional interaction of spatio-temporal information. As shown in Fig. 5(c), the temporal and spatial injection sub-modules are formulated as

$$u_s = U_S(S_t^{n-1,k}) = \sigma(W_s * S_t^{n-1,k}), \quad (18)$$

$$S_t^{n,k} = u_s \odot S_t^{n-1,k} + (1 - u_s) \odot T_{est}, \quad (19)$$

$$u_t = U_T(T_{est}) = \sigma(W_t * T_{est}), \quad (20)$$

$$T_t^{n,k} = u_t \odot T_{est} + (1 - u_t) \odot S_t^{n-1,k}, \quad (21)$$

where $u_s$ and $u_t$ are ratios generated from $U_S$ and $U_T$, guiding the injection of information form temporal to spatial domain and its reversal. This constrains the output to maintain consistency between two dimension.

## 4 EXPERIMENTAL RESULTS

### 4.1 Experimental Settings

We select $K = 4$ MTFs in ICP methods with $TRF = (7, 9, 11, 13)$. MSTAUs are stacked with $N = 4$ layers, each of which takes $\tau = 5$ historical states as observations. We consider **scene reconstruction** and **object classification** as representatives of spike-based visual applications at distinct semantic levels. The former is validated on *S-VIMEO* dataset [9], while the latter is evaluated on *S-MNIST*, *S-CIFAR*, and *S-CALTECH* datasets [41]. For performance

| Predicted Sequence | Processing Method | t+1 | t+3 | t+5 | t+10 |
|---|---|---|---|---|---|
| $\{\hat{F}_t\}$ | PredRNN w/o ICP | 8.31 | 11.74 | 14.82 | 18.05 |
| | MAU w/o ICP | 7.70 | 10.97 | 14.03 | 16.78 |
| | PredRNN w/ ICP | 7.56 | 9.97 | 12.10 | 16.37 |
| | MAU w/ ICP | 6.89 | 9.29 | 11.61 | 15.99 |
| | **MSTAU** | **6.26** | **8.61** | **11.04** | **15.34** |
| $\{\hat{f}_t^1\}$ | PredRNN w/ ICP | 6.59 | 9.61 | 12.18 | 16.62 |
| | MAU w/ ICP | 6.11 | 8.62 | 10.63 | 14.66 |
| | **MSTAU** | **6.03** | **8.47** | **10.22** | **13.98** |
| $\{\hat{f}_t^2\}$ | PredRNN w/ ICP | 4.32 | 6.51 | 7.55 | 9.17 |
| | MAU w/ ICP | 4.04 | 5.56 | 6.94 | 8.71 |
| | **MSTAU** | **3.96** | **5.34** | **6.59** | **8.10** |
| $\{\hat{f}_t^3\}$ | PredRNN w/ ICP | 2.71 | 4.24 | 5.71 | 6.65 |
| | MAU w/ ICP | 2.57 | 2.94 | 5.10 | 6.16 |
| | **MSTAU** | **2.47** | **2.77** | **4.80** | **5.67** |
| $\{\hat{f}_t^4\}$ | PredRNN w/ ICP | 1.12 | 1.85 | 2.51 | 5.12 |
| | MAU w/ ICP | 1.01 | 1.71 | 2.22 | 4.10 |
| | **MSTAU** | **1.01** | **1.63** | **2.08** | **3.83** |

Table 2: Comparison of prediction error measured by MSE (in units of 1e-4) for various spike processing methods on the composed representation and features at different scales. Results demonstrate that introducing intra-scale correlation and further incorporating inter-scale one both lead to significant enhancements in precision.

| Processing Method | PredRNN [30] | | MAU [6] | | **MSTAU** |
|---|---|---|---|---|---|
| | w/o ICP | w/ ICP | w/o ICP | w/ ICP | |
| t+1 | 32.67 | 32.93 | 33.36 | 33.73 | **34.42** |
| t+3 | 30.86 | 31.27 | 31.74 | 32.51 | **33.60** |
| t+5 | 29.02 | 29.82 | 29.86 | 31.04 | **32.18** |
| t+10 | 25.24 | 26.20 | 27.44 | 28.86 | **30.01** |

Table 3: Comparison of PSNR performance for various spike processing methods in scene reconstruction tasks. Results show that performance is greatly improved through incorporating ICP method for different backbones. MSTAU further outperforms other approaches by introducing multi-scale aggregation.

comparison, we utilize two state-of-the-art (SOTA) temporal prediction models, PredRNN[30] and MAU[6], as backbones to assess the reconstruction precision and classification accuracy. [2]

## 4.2 Reconstruction Oriented Prediction

**Effectiveness of intra-scale correlation.** Through comparing the predictive accuracy with and without incorporating ICP method, we validate the effectiveness of intra-scale correlation in guiding spike prediction. Without ICP method, the predictor forecasts representations for future moments directly, corresponding to Eq. 1, 2 and 3. In contrast, with ICP method composed of Eq. 8, 10, 11 and 3, the predictor forecasts multi-scale features, which

---

[2]Details are illustrated in the supplementary material.

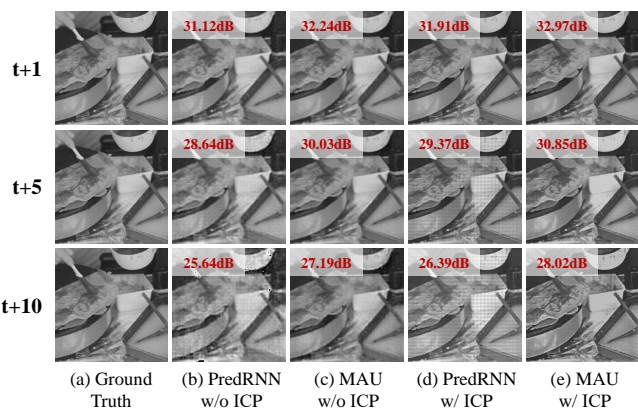

| | (a) Ground Truth | (b) PredRNN w/o ICP | (c) MAU w/o ICP | (d) PredRNN w/ ICP | (e) MAU w/ ICP |

Figure 6: Visual comparison between scene reconstructed through feature-domain approaches with and without ICP method. Zoom for better visibility.

are then composed as visual representations to support intelligent applications. As shown in Table. 2, the prediction error measured by MSE on representation $\{\hat{F}_t\}$ decreases by 7.3% for PredRNN and 8.9% for MAU when the ICP method is applied. This improvement is attributed to the ICP method's capability to effectively utilize intra-scale correlations to predict spatio-temporal contents with high fidelity. The enhanced accuracy is evident when comparing the prediction errors of $\{\hat{f}_t^{\{1:4\}}\}$ and $\{\hat{F}_t\}$ in Table. 2, demonstrating the more complex spatio-temporal characteristics of $F_t$ make it more challenging to predict accurately compared to $f_t^{\{1:4\}}$, which is consistent with the conclusion in Fig. 3(b). ICP method, by composing accurate $\hat{f}_t^{\{1:4\}}$ as $\hat{F}_t$, simplifies a complex prediction into multiple sub-predictions and reaches high precision through the strong relevance within each scale.

Precise scenes can be reconstructed subsequently from this accurate predicted representation, with statistical results shown in Table. 3, demonstrating the efficacy of intra-scale correlation for visual perception. Compared to the absence of ICP method, employing ICP method leads to an average increase of 2.7% in the PSNR, with the gain growing as the prediction time interval extends (0.32dB, 0.59dB, 0.99dB, 1.19dB). This suggests that ICP can exploit the stronger temporal continuity of features at different scales, resulting in more significant performance enhancements for long-term predictions of scene sequences. We also visualize reconstructed scenes based on predicted features through ICP method, as shown in Fig.6. Without ICP method, local low-intensity features are often misinterpreted as noise, causing uncontrolled blurring in the reconstructed content. Moreover, these noisy features tend to propagate over time, leading to accumulation of blurring and other artifacts that significantly deteriorate scene quality. On the other hand, implementing the ICP method stabilizes the predicted features, preventing high-frequency information from being overshadowed by low-frequency noise. Consequently, scenes reconstructed with ICP exhibit distinct edges and rich texture details.

**Effectiveness of inter-scale correlation.** We demonstrate the significant enhancement in perception and prediction achieved through further incorporation of inter-scale correlations by comparing MSTAU with other ICP methods. According to Table. 2,

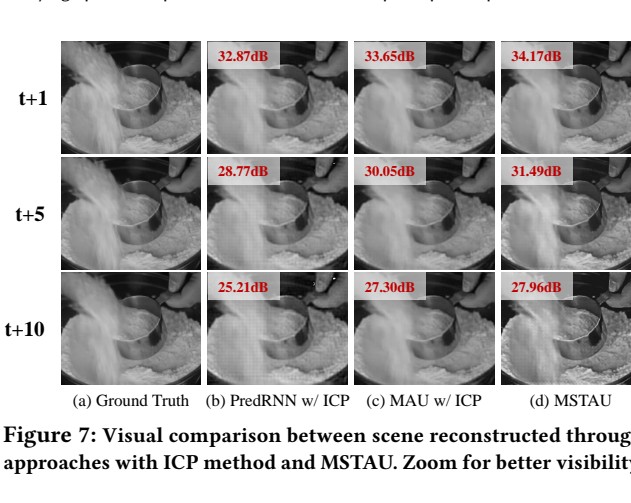

Figure 7: Visual comparison between scene reconstructed through approaches with ICP method and MSTAU. Zoom for better visibility.

the MSTAU improves accuracy of each feature sequence by 4.9%, 7.5%, 8.6% and 7.0% compared to ICP methods, which is consistent with the PLCC in Table. 1 approximately. Feature corresponding to TRF=11 exhibit the strongest correlation with others and guiding it with other features can therefore maximize the prediction accuracy of $\hat{f}_t^3$, while $\hat{f}_t^1$ corresponding to TRF=7 demonstrates opposite characteristics. This indicates that content with stronger correlations to features at other scales can be guided by inter-scale information more effectively, resulting in more precise predictions. Moreover, the accuracy of predicted representation $\hat{F}_t$ can also improved with assistance of MSTAU, as illustrating in Table. 2. Compared to ICP approaches only considering intra-scale correlations, MSTAU enhances the precision of composed representations by an average of over 8.9%. This improvement exhibits a roughly decreasing trend with the increase in prediction temporal interval (13.2%, 10.5%, 6.8%, 5.2%). Although features are correlated, the spatio-temporal characteristics corresponding to different TRFs are distinct essentially. As a result, the correlation between contents within different features significantly decreases over time, and the temporal estimation generated with the help of prior knowledge shows noticeable distortion. These factors collectively contribute to the decrease in accuracy gain.

For visualization, we showcase the predictive reconstruction scenes of MSTAU in Fig. 7. MSTAU outperforms methods with no inter-scale correlation by predicting and generating higher quality scenes in short-term scenarios (*e.g.* the spoon with more distinct edges and the flour with a grainier texture). However, we note that as the prediction interval extends, the quality of the reconstructed scenes deteriorates, characterized by overly high local contrast and a noticeable light-dark grid effect (Fig. 7(d)). This decline in quality stems from the decreasing correlation of features across different scales, resulting in content inconsistencies and misalignments. The reliance on the pixel-wise L2-norm as the only constraint causes the model to overlook the rationality of local textures and their correlation with adjacent frames during processing. Future work can address this issue by incorporating inter-frame constraints.

## 4.3 Classification Oriented Prediction

**Effectiveness of inter-scale correlation.** Compared to ICP approaches, MSTAU achieves an improvement in classification accuracy of 0.70%, 2.20%, and 1.47% for the *S-MNIST*, *S-CIFAR* and

| Dataset | Processing Method | t+1 | t+3 | t+5 | t+10 |
|---|---|---|---|---|---|
| *S-MNIST* | PredRNN w/ ICP | 98.64 | 97.85 | 97.02 | 94.40 |
| | MAU w/ ICP | 98.86 | 98.08 | 97.18 | 94.96 |
| | **MSTAU** | **99.13** | **98.77** | **98.06** | **95.92** |
| *S-CIFAR* | PredRNN w/ ICP | 67.10 | 64.97 | 63.24 | 57.89 |
| | MAU w/ ICP | 67.55 | 65.68 | 63.95 | 58.07 |
| | **MSTAU** | **68.02** | **67.43** | **66.28** | **62.30** |
| *S-CALTECH* | PredRNN w/ ICP | 76.45 | 75.32 | 73.97 | 69.01 |
| | MAU w/ ICP | 76.71 | 75.77 | 74.60 | 69.85 |
| | **MSTAU** | **77.25** | **76.84** | **75.76** | **72.97** |

Table 4: Comparison of Top-1 classification accuracy for various spike processing methods across different datasets. Results indicate that incorporating intra-scale and inter-scale correlations can achieve more precise categorization, and the precision remains stable over longer prediction intervals.

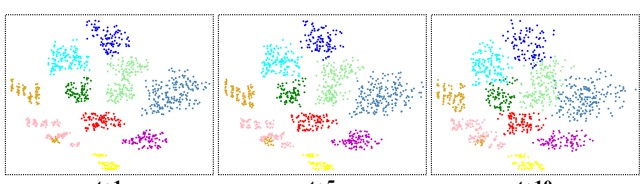

Figure 8: Visualization of T-SNE on *S-MNIST* through MSTAU. Results indicate that samples of the same category cluster closely despite of large prediction intervals, making the classification precise.

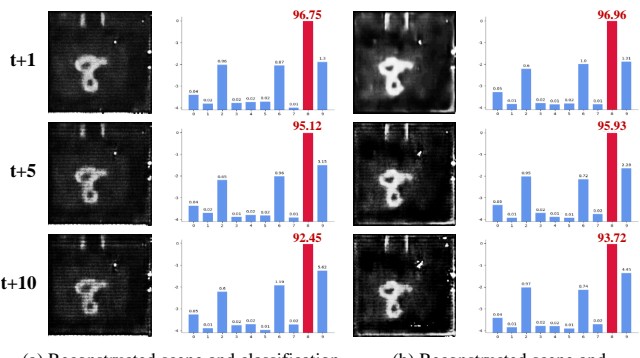

(a) Reconstructed scene and classification confidence via MAU w/ ICP

(b) Reconstructed scene and classification confidence via MSTAU

Figure 9: Reconstructed scenes and classification confidence through MSTAU and MAU with ICP on *S-MNIST*. Results demonstrate that compact representations can simultaneously accomplish multiple visual applications with different semantic levels.

*S-CALTECH* respectively as shown in Table. 4. This demonstrates that introducing inter-scale correlation for prediction can enhance the semantic fidelity of visual representations, providing support for spike-based intelligent applications in perception and understanding. Additionally, we visualize the last layer of latent features from *S-MNIST* via t-Distributed Stochastic Neighbor Embedding (t-SNE) [29], as shown in Fig. 8. Results indicate that predicted labels for samples within the same category remain tightly clustered in the high-dimensional space, thereby ensuring high fidelity in classification accuracy. However, the degree of congregation

| Processing Method | t+1 | t+3 | t+5 | t+10 |
|---|---|---|---|---|
| PredRNN [30] | 31.72 | 29.23 | 27.42 | 23.28 |
| MAU [6] | 32.27 | 30.02 | 28.16 | 25.79 |

**Table 5: PSNR performance for reconstruction and prediction in scene-domain. Results indicate that pixel-wise distortion accumulate over time, leading to extremely poor predictions.**

slightly decreases over time, leading to a small number of samples being misclassified and a slight reduction in classification precision essentially.

**Versatility for multiple tasks.** We further validate the versatility and robustness of our method for multiple intelligent tasks on *S-MNIST*. Fig. 9 shows the results processed by the reconstruction and classification network respectively from the predicted visual representation $\hat{F}_t$. The proposed ICP method and MSTAU can reconstruct precise scenes while ensuring accurate classification with high confidence. Moreover, we note that although the distortion in reconstructed scenes becomes more noticeable as the prediction interval extends, the classification accuracy consistently remains high. This indicates that for visual tasks at different semantic levels, performance exhibits varying levels of temporal decay with the prediction interval, suggesting that our method can be paired with different analysis models to achieve diverse intelligent applications with varying granularity and precision requirements.

### 4.4 Ablation Study

**Effectiveness of feature-domain processing.** We take scene reconstruction to demonstrate the rationality of feature-domain processing compared to that in scene-domain. Comparison of results in Table. 3 and 5 show that feature-domain methods outperform scene-domain ones by 8.4% and 6.4% for PredRNN and MAU respectively. The average PSNR gain of the reconstructed scenes is positively proportional to the prediction interval (1.0dB, 1.7dB, 1.7dB, 1.8dB), indicating that visual features have better temporal coherence and contain more abundant spatio-temporal information to ensure efficient prediction and authentic reconstruction compared to reconstructed scenes. As shown in Fig. 1, results generated through feature-domain processing broadly eliminate content-irrelevant artifacts and blurring, while retain high-frequency texture details in the meanwhile . This indicates that visual features contain comprehensive information of various semantic granularities, which is utilized for reconstruction and prediction with higher accuracy.

**Effectiveness of multi-scale aggregation.** We verify the rationality of proposed multi-scale aggregation through ablation experiments. In the aggregation process described as Eq.14 and 15, the inter-scale correlations between spatial states are utilized to obtain the weight $r_i^j$, which controls the aggregation of temporal state from scale $i$ to $j$. The weight matrix $[r_i^j]$ is visualized in Fig.10. The distribution of $[r_i^j]$ resembles the PLCC distribution in Table. 1, suggesting that features with stronger correlations are more effectively guided and aggregated through information interaction, leading to enhanced prediction accuracy.

Additionally, we further confirm the impact of aggregation process on the spatial and temporal states of the output, as shown in Fig. 11. $u_f$, $u_s$, and $u_t$ correspond to Eq. 16, 18 and 20 respectively.

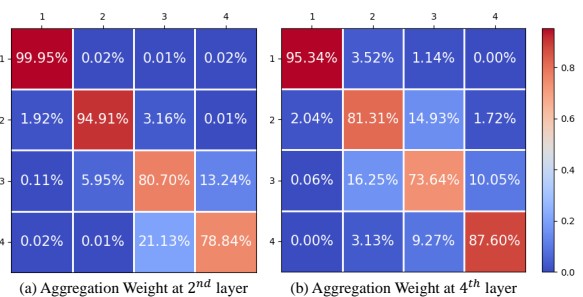

(a) Aggregation Weight at $2^{nd}$ layer      (b) Aggregation Weight at $4^{th}$ layer

**Figure 10: Visualization of the aggregation weight matrices for each layer of MSTAU. Results reveal that features with higher correlations are more effectively guided and aggregated through information interaction.**

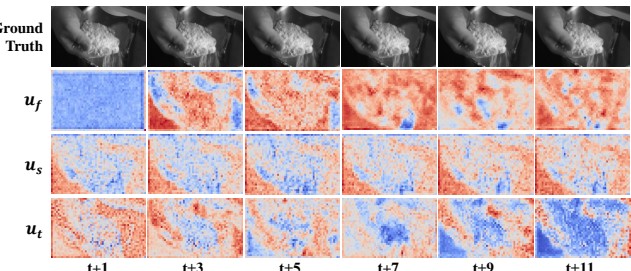

**Figure 11: Visualization of the fusion and injection ratio within MSTAU. Results indicate that as the estimations and aggregations stabilize, the weights allocated to fusion and bidirectional injection gradually increase, demonstrating the effectiveness of spatio-temporal aggregation.**

The sequence of $u_f$ indicates that in the initial stage, the information within $T_{att}^n$ is not yet accurate, so the aggregation module obtain most content from $T_{t-1}^{n,k}$. As time accumulates, $T_{att}^n$ gradually stabilizes. For stationary areas, the module tends to obtain information from long-term estimates, while only whose with intense motion aggregate momentary information. Similarly, sequences of $u_s$ and $u_t$ indicate that $T_{agg}$ obtained mainly from momentary content has a significant deviation from $S_t^{n-1,k}$, making it difficult to fuse and obtain spatio-temporal states with high accuracy. As the aggregation quality improves over time, the fusion module can gradually align $S_t^{n-1,k}$ and $T_{agg}$ on content, thereby jointly generating $S_t^{n,k}$ and $T_t^{n,k}$ with high precision. This provides support for subsequent visual applications at various semantic granularities.

## 5 CONCLUSION

We propose an innovative spike processing method for visual perception and temporal prediction utilizing compact spike representation with high versatility, providing a fresh perspective on spike intelligence. Taking multi-scale correlation into consideration, our method effectively models spatio-temporal information at different scales, facilitating proactive decision-making in scenarios with diverse semantic complexities. Experimental results confirm that our method significantly enhance predictive accuracy of spike perception, propelling the widespread adoption and application of spike cameras and further contributing for a tech-harmonious world.

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
