# OpenReview forum: "Unifying Spike Perception and Prediction: A Compact Spike Representation Model using Multi-scale Correlation"
_acmmm.org/ACMMM/2024/Conference — MM2024 Poster_

### Official Review · Reviewer_PuPi · 2024-05-24

**Rating:** 2
**Confidence:** 3

**Summary:**

The paper introduces an innovative spike processing method, integrating perception and prediction using compact spike representations through multi-scale correlations. The proposed method, utilizing intra-scale correlation based prediction (ICP) and a Multi-scale Spatio-Temporal Aggregation Unit (MSTAU), aims to enhance predictive accuracy for visual tasks like scene reconstruction and object classification. The experiments demonstrate significant improvements in predictive accuracy, scene reconstruction quality, and classification accuracy, providing a robust foundation for spike processing applications.

**Strengths:**

Novel Methodology: The introduction of ICP and MSTAU to leverage both intra-scale and inter-scale correlations offers a fresh perspective on spike processing. This approach enhances the precision of temporal predictions and visual perception tasks.
Experimental Validation: The paper provides extensive experimental results showcasing improvements in predictive accuracy, scene reconstruction quality (3.49dB increase in PSNR), and classification accuracy (2.20% improvement in Top-1 accuracy).
Versatility: The proposed method is adaptable to both low-level and high-level spike visual tasks, indicating its potential applicability across a wide range of intelligent visual applications.
Comprehensive Analysis: The paper includes a thorough analysis of spatio-temporal correlations and the effectiveness of the proposed modules, supported by detailed experimental results and visual comparisons.

**Limitations:**

The authors mentioned that the code is available in code.py, but it is currently an empty file.
The description of the implementation details, especially for the multi-scale spatio-temporal aggregation unit, might be too complex for straightforward replication without an accompanying detailed pseudocode or source code. The mention of code.py being empty is a significant oversight in this respect.
While the paper mentions using the S-MNIST, S-CIFAR, and S-CALTECH datasets for experiments, it lacks detailed descriptions of the specific experimental settings and data preprocessing. This may cause difficulties for readers trying to replicate the experiments.
The paper selects PredRNN and MAU as comparison methods but does not provide detailed reasons for choosing these methods. It also does not cover other possible comparison methods, such as other models based on spatio-temporal correlations.

**Suitability:**

1

---

### Official Review · Reviewer_mbPG · 2024-05-24

**Rating:** 5
**Confidence:** 2

**Summary:**

This paper presents a novel method for spike signal processing that aims to unify spike perception and prediction through a compact spike representation model using multi-scale correlation. The introduction critiques existing methods for their lack of simultaneous visual perception and temporal prediction capabilities, and proposes a multi-scale spatio-temporal aggregation unit (MSTAU) to address these deficiencies. The method exploits intra-scale and inter-scale correlations to increase the efficiency of spike signal processing, demonstrating significant improvements in scene reconstruction and object classification. Specifically, the method achieved a 3.49dB increase in scene reconstruction quality and a 2.20% increase in classification accuracy.

**Strengths:**

The approach proposed in this paper is novel. The authors develop a new way of processing spike signals, the Multiscale Spatio-Temporal Aggregation Unit (MSTAU), by analyzing the spike signals and spatial and temporal correlations on different time scales.

**Limitations:**

1. Does MSTAU incur higher computational costs? It is recommended that the authors provide more details on the computational cost and practicality of deploying this method in real applications.
2. An analysis of the limitations of existing methods is lacking in the descriptions in the related work in Section 2.
3. Many symbols in the main text lack corresponding meanings, which causes reading problems, although the appendix provides a table of symbol descriptions.

**Suitability:**

2

---

### Official Review · Reviewer_QQT9 · 2024-05-24

**Rating:** 5
**Confidence:** 3

**Summary:**

Bio-inspired cameras have a wide range of applications, however, traditional methods are difficult to provide both visual perception and temporal prediction. In this paper, the authors propose a method for spike representation using intra-scale correlation by analyzing the spatio-temporal correlation of spike information at different time scales, in addition to which, the authors propose multi-scale spatio-temporal aggregation units to further achieve efficient perception and accurate prediction. The authors conducted relevant experiments on scene reconstruction and object classification to achieve higher network model performance.

**Strengths:**

1. The paper is clearly motivated. The authors first summarize two problems with existing methods for simultaneous perception and prediction of impulse signals when applied to visual tasks, namely lack of generality and prediction distortion. Among them, prediction distortion is not only a lack of pixel-level accuracy, but also aggravates the distortion over time. Therefore, there is an urgent need to propose an effective pulse processing method with both sensing and prediction functions, which is the contribution of this paper.
2. The authors propose a spike processing method for multi-scale correlation, which is divided into intra- and inter-scale correlation. The method is able to predict intra-peak information with higher efficiency and fidelity in the time domain compared to previous methods, which further provides better performance for downstream applications.
3. The experiments are adequate and convincing. The authors compare with SOTA in scene reconstruction and object classification. I think the ablation experiment part is commendable and I clearly see the effectiveness of the two proposed methods, Intra-scale Correlation based Prediction and Multi-scale Spatio-Temporal Aggregation Unit.

**Limitations:**

I see no major flaws in this paper and I am inclined to accept it.

I believe that the method is valid, nevertheless, I am not sure about the extent to which it contributes to the field as a whole, i.e. how widely it is applied and how much it contributes to the field as a whole. I therefore need a reviewer who is more experienced in this area to help me make a judgement. My rating at this stage is ‘Weak Accept’.

**Suitability:**

2

---

### Official Review · Reviewer_YFin · 2024-05-24

**Rating:** 4
**Confidence:** 3

**Summary:**

The paper addresses the functional requirements of bio-inspired cameras in extreme scenarios by proposing a novel spike signal processing method. The paper analyzes the spatio-temporal correlations of spike information across different time scales and introduces an innovative approach to spike processing. This method leverages intra-scale correlations to achieve a more compact representation of spikes and enhance predictive accuracy. Furthermore, the paper proposes a Multi-scale Spatio-Temporal Aggregation Unit (MSTAU) that further capitalizes on inter-scale correlations to realize efficient perception and precise prediction.

Experimental results demonstrate that the proposed method has achieved significant improvements in scene reconstruction and target classification. The method can adapt to different visual applications by switching analysis models, providing a new perspective for spike signal processing.

**Strengths:**

The MSTAU proposed in the paper is a novel innovation, and the paper analyzes its theoretical rationality and demonstrates its advantages through experiments; The paper thoroughly demonstrates the effectiveness of the proposed spike processing method through extensive experiments, and the visualization of multiple experimental results more intuitively showcases the efficacy of the proposed approach.

**Limitations:**

Can an analysis of the algorithmic complexity be added to evaluate the efficiency of the algorithm? Can the effectiveness of the innovation be further demonstrated by adding tests on other datasets and tasks in terms of multi-tasking? Can you provide the specific meanings of hyperparameters in experimental settings and the reasons for determining the values of hyperparameters?

**Suitability:**

2

---

### Meta-Review · Area_Chair_uEGR · 2024-07-01

**Recommendation:** Accept (Poster)
**Confidence:** 5

**Metareview:**

This paper introduces an innovative approach to processing spike signals from bio-inspired cameras, addressing challenges posed by the intricate nature of these signals. The authors analyze spatio-temporal correlations at different temporal scales and propose a new spike processing method for compact spike representations, utilizing intra-scale correlation for higher predictive accuracy. Additionally, they present a multi-scale spatio-temporal aggregation unit (MSTAU) that leverages inter-scale correlation for efficient perception and precise prediction. Experimental results show significant improvements in scene reconstruction quality and object classification accuracy, with increases of 3.49dB and 2.20%, respectively. The proposed method also accommodates various visual applications by switching analysis models, offering a novel perspective on spike processing. The proposed method for spike signal processing represents a significant advancement in the field of bio-inspired cameras and intelligent applications. The innovative use of multi-scale spatio-temporal correlations leads to notable improvements in performance metrics, while the flexibility of switching analysis models adds to its practical value. The thorough experimental evaluation provides strong evidence of the method's effectiveness. Although there are minor areas for improvement in explaining algorithmic complexity and discussing hardware dependence, these do not detract from the overall contribution and impact of the work.